# Non-Isothermal Crystallization Kinetics of Poly(Ethylene Glycol)–Poly(l-Lactide) Diblock Copolymer and Poly(Ethylene Glycol) Homopolymer via Fast-Scan Chip-Calorimeter

**DOI:** 10.3390/polym13071156

**Published:** 2021-04-04

**Authors:** Dejia Chen, Lisha Lei, Meishuai Zou, Xiaodong Li

**Affiliations:** School of Materials Science and Engineering, Beijing Institute of Technology, Beijing 100081, China; cdj2021@126.com (D.C.); 3220201147@bit.edu.cn (L.L.); zoums@bit.edu.cn (M.Z.)

**Keywords:** fast-scan chip-calorimeter, non-isothermal crystallization kinetics, poly(ethylene glycol)–poly(l-lactide), diblock copolymer, double-crystallizable, fast cooling rate

## Abstract

The non-isothermal crystallization kinetics of double-crystallizable poly(ethylene glycol)–poly(l-lactide) diblock copolymer (PEG-PLLA) and poly(ethylene glycol) homopolymer (PEG) were studied using the fast cooling rate provided by a Fast-Scan Chip-Calorimeter (FSC). The experimental data were analyzed by the Ozawa method and the Kissinger equation. Additionally, the total crystallization rate was represented by crystallization half time *t*_1/2_. The Ozawa method is a perfect success because secondary crystallization is inhibited by using fast cooling rate. The first crystallized PLLA block provides nucleation sites for the crystallization of PEG block and thus promotes the crystallization of the PEG block, which can be regarded as heterogeneous nucleation to a certain extent, while the method of the PEG block and PLLA block crystallized together corresponds to a one-dimensional growth, which reflects that there is a certain separation between the crystallization regions of the PLLA block and PEG block. Although crystallization of the PLLA block provides heterogeneous nucleation conditions for PEG block to a certain extent, it does not shorten the time of the whole crystallization process because of the complexity of the whole crystallization process including nucleation and growth.

## 1. Introduction

Poly(ethylene glycol)–poly(l-lactide) diblock copolymer (PEG-PLLA) is widely used in the medical field due to its biodegradability, blood compatibility and drug permeability [1]. In recent years, a lot of literature has reported the application of PEG-PLLA with different structures in the field of medicine, and pointed out its disadvantages such as high crystallinity, slow degradation rate and being insoluble in water [2,3,4]. In fact, the crystal structure of PEG-PLLA will affect its thermal, mechanical and other physical properties, thus determining the application performance [5,6]. Understanding and controlling the crystallization behavior have provided an effective method for predicting and adjusting the physical properties of crystalline polymers. In addition, just as the additive nanoparticles have an influence on the crystallization of nanocomposite polymers, the crystallization of different blocks in the double-crystallizable block copolymer PEG-PLLA is bound to influence each element, which is worth exploring [7,8,9].

In practice, semi-crystalline polymer materials are usually non-isothermal crystallization during the processing and cooling process, which largely determines the ultimate properties and practical application value of the materials. Therefore, it is of great significance to study the crystallization process of polymers under non-isothermal conditions for the technological optimization and preparation of high-performance polymer materials. Yang, J.L. et al. [10] and Kong, X.H. et al. [11] studied the non-isothermal crystallization process of PEG-b-PLLA and PET-b-PEO (poly(ethylene terephthalate)-poly(ethylene oxide) diblock copolymer) by conventional differential scanning calorimetry (DSC) as a conventional means to study the polymer crystallization kinetics. However, the fastest cooling rate of conventional DSC usually cannot inhibit the crystallization of samples. It is difficult to avoid crystal nucleation and growth during cooling process, which affects the nucleation behavior of polymer crystals in low temperature regions [12,13,14,15]. The actual cooling rate in machining is often greater than the maximum cooling rate that can be achieved by conventional DSC, while it can be achieved by Fast-Scan Chip-Calorimetry (FSC) because of extremely fast heating and cooling rates. Although FSC is widely used in homopolymer, such as isotatic polypropylene (*i*PP) [16], polyetheretherketone (PEEK) [17], poly(ε-caprolactone) (PCL) [18], etc., few people apply it to copolymers, let alone double-crystallizable block copolymers, because of the complexity of the crystallization process. Due to the complex crystallization behavior of double-crystallizable block copolymers, the immaturity of the non-isothermal crystallization kinetics theory and the lack of precise and appropriate instruments and methods, there are few studies on the non-isothermal crystallization kinetics of PEG-PLLA. However, the crystallizing temperature difference between PEG block and PLLA block in PEG-PLLA is large, which can make PLLA crystallize in a large range while PEG is in the molten state [19]. This feature provides a good starting point for us to study its non-isothermal crystallization kinetics. Therefore, we innovatively proposed a new method to explore the non-isothermal crystallization kinetics of the double-crystallizable block copolymer PEG-PLLA via FSC, which is expected to be applied in a wider field.

In this paper, two cooling methods are used to cool rapidly PEG-PLLA by FSC, and the Ozawa method [20] and Kissinger equation [21] are applied to PEG-PLLA diblock copolymer to explore its non-isothermal crystallization kinetics and compare them with those of the PEG homopolymer.

## 2. Materials and Methods

### 2.1. Materials

Crystallization experiments were performed with PEG-PLLA and PEG, which were provided by the Shenzhen Meiluo Technology Co., Ltd. (Guangdong, China). PEG has a weight-average molecular weight of *M*_w_ = 5000 g/mol. The weight-average molecular weight of the PEG block and PLLA block in the PEG-PLLA are 5000 g/mol and 2000 g/mol, respectively, and the weight percentages for each block are 71.42% and 28.57%, respectively.

### 2.2. Test Instrument

A chip-calorimeter instrument, Flash DSC 2+ (made by Mettler-Toledo, Zurich, Switzerland) equipped with a Huber TC-100 intracooler was employed. The maximum heating rate and the maximum cooling rate can reach 50,000 K/s and 40,000 K/s, respectively. FSC has been widely used due to its advantages such as ultra-fast heating and cooling scanning rate, outstanding temperature control ability and accurate time resolution. It can inhibit the crystallization nucleation of the polymer sample during the process of heating and cooling, and avoid the influence on the subsequent crystallization kinetics test. The results enable us to obtain the initial information about the structure of the aggregated states inside the polymer crystals, to produce a new understanding of the annealing and melting behavior of polymer crystals, and to deepen understanding of the nucleation mechanism of polymer crystals. Schick, C. et al. [22,23,24] have suggested that heterogeneous nucleation can be effectively bypassed and homogeneous nucleation can be observed in the bulk at low temperatures, using the fast cooling rate provided by FSC. Based on the proposal of Schick, C. et al., this study assumes that this hypothesis is valid; therefore, the crystallization kinetics of diblock copolymer can be deduced.

### 2.3. Methods

The sample was prepared under an optical microscope and then transferred onto the cross-marked area of the sample cell. A pre-melting operation was adopted to ensure good thermal contact between the sample and sensor before applying the preset temperature program. In this step, the sample was heated to 180 °C and then cooled to room temperature; both heating and cooling were at the rate scale of 1 K s^−1^. In order to provide comparable data obtained at the scanning rates of different orders of magnitude, each heat flow rate was normalized with the relative scanning rate, and the heat flow measured by Flash DSC 2+ was transformed into apparent heat capacity. In our measurements, the same samples were used on the same sensor, and the results shared the same baseline. In addition, since different cooling rates employed the same sample and the results were independent of sample mass, this parameter was not essential in the present analysis. The data treatment was performed by STAR^e^ software (Version 10.0).

FSC can provide an extremely fast cooling rate that inhibits any ordered process, i.e., crystallization. The rate at which cooling crystallization is precisely inhibited is called the critical cooling rate. Thus, the crystallization of the PLLA block can be inhibited when the cooling rate is higher than the critical rate to 60 °C [14], so that the PLLA block crystallized together with the PEG block in the process from 60 °C to −80 °C. Therefore, we used two different cooling methods to study PEG-PLLA (methods 1 and 2) and used method 1 for PEG. Method 1: cool directly from 180 °C to −80 °C, as shown in Figure 1a; Method 2: cool from 180 °C to 60 °C, cooling to −80 °C, respectively, by more than the critical cooling rate (2000 K/s) and cooling rate for studying, as shown in Figure 1b. The heating rate is 50 K/s, isothermal at 180 °C for 1 s to erase the thermal history.

## 3. Results and Discussion

### 3.1. Analysis of Non-Isothermal Crystallization Curves

Generally, the crystallinity of polymers is relatively low, so it is not convenient to describe the crystallization process, so a more reasonable relative scale was adopted. The relative crystallinity *X*_C_ was characterized by the heat enthalpy method [25]:(1)$$X_{c}=\frac{\int_{T0}^{Tc}\left(\frac{dH}{dT}\right)dT}{\int_{T0}^{T\infty }\left(\frac{dH}{dT}\right)dT}$$
where *T*_0_, *T*_c_ and *T*_∞_ are the initial, final, ultimate crystallization temperatures, respectively, and *H* is the crystallization enthalpy.

The results of the non-isothermal crystallization behavior of PEG and PEG-PLLA are shown in Figure 2. At relatively low cooling rates, Figure 2a only has one crystallization peak around −25 °C, while Figure 2b has two crystallization peaks around 75 °C and −25 °C, respectively. The obvious crystallization peak near 75 °C in Figure 2b was attributed to the crystallization peak of the PLLA block [16]. Thus, it can be concluded that the PEG-PLLA copolymer is a double-crystallizable block copolymer, and the crystallization temperature of the two blocks is a big difference about 105 °C. In Figure 2b, the crystallization peak near 75 °C is very flat at 100 K/s; however, it becomes more and more convex with the increase in cooling rate, until it disappears at 2000 K/s because crystallization is inhibited. In Figure 2c, crystallization in the PLLA block is inhibited between 180 °C and 60 °C due to the ultra-fast cooling rate (2000 K/s). Therefore, there are no crystallization peaks in this temperature range. 

With the increase in cooling rate, the crystallization peaks of the PEG block in PEG-PLLA and PEG tend to flatten during the crystallization process, which is due to spherulite impingement and crowding [26]. The crystallization peaks above all become wider and shift to the lower temperature region with the increase in cooling rate; that is, the faster the cooling rate is, the later the crystallization time is. This is because the cooling rate is too fast for the polymer to form a crystal within the limited time. Moreover, it can be known that the critical cooling rates of (a), (b) and (c) are 1000–1200 K/s, 1200–1600 K/s and 1000–1200 K/s, respectively.

The starting temperature (*T*_s_), temperature at maximum heat flow (*T*_max_), and corresponding relative crystallinity *X*_max_ under different cooling rates of PEG and PEG-PLLA are listed in Table 1. For one thing, at each cooling rate, *T*_s_ and *T*_max_ are the maximum of b and the minimum of a, and both decrease with the increase in cooling rate. Compared with PEG, the crystallization peaks of the PEG block in the PEG-PLLA move towards the higher temperature region known from increase in *T*_s_ and *T*_max_. Compared with method 1, the crystallization peaks of the PEG block in the PEG-PLLA using method 2 move towards the higher temperature region known from increase in *T*s and *T*_max_. On the other hand, at the same cooling rate, *X*_max_ in (b) is higher than that in (a). It can be understood that the PEG block in PEG-PLLA has better crystallizability because the crystallized PLLA block promotes the ability of the PEG molecular chain to diffuse to the crystallization site. Additionally, the average peak relative crystallinity of (b) is greater than that of (c), which means that the crystallization of PEG block on the basis of block PLLA crystallization is easier than that of the PEG block and PLLA block crystallized together; that is, the first crystallized PLLA block promotes the crystallization of the later PEG block. The above quantitative data indicate that the non-isothermal crystallization kinetics of PEG and the PEG block in PEG-PLLA are different, and the two cooling methods have different effects on the crystallization of the PEG block. When the amorphous phase is cooled to a low temperature, the PLLA block in (b) crystallizes first and grows freely from the homogeneous melt to form a specific morphology, which must have an effect on the crystallization of the PEG block later.

### 3.2. Analysis by the Ozawa Equation

Non-isothermal crystallization kinetics were analyzed by the Ozawa method [20]. Ozawa assumed that the non-isothermal crystallization process could be composed of infinitesimal isothermal crystallization steps and extended the Avrami equation [27] to the non-isothermal conditions. The Ozawa equation for calculating the relative crystallinity at temperature *T* as follows:(2)$$1-X_{c}=\exp\left[-\frac{K\left(T\right)}{a^{q}}\right]$$
where K(*T*) is a cooling function of the process, *a* is the cooling rate, and *q* is the Ozawa exponent that relies on the crystal growth and nucleation mechanism, corresponding to one-dimensional growth, *q* = 2; two-dimensional growth, *q* = 3; three-dimensional growth, *q* = 4. Being expressed in a double logarithmic form, Equation (2) can be further rewritten as
(3)$$\ln[-\ln\left(1-X_{c})\right]=\mathrm{lnK}\left(T\right)-q\cdot \ln a$$

In the actual measurement, the relative crystallinity was selected at a given temperature *T* and different cooling rates from the non-isothermal crystallization curves. According to Equation (3), the results of the Ozawa analysis are shown in Figure 3 by plotting ln[−ln(1−*X*_c_)] versus ln*a* for crystallization temperature at −19 °C, −20 °C, −21 °C and −22 °C. Then, K(*T*) and q are determined by the intercept and slope, respectively. The results are summarized in Table 2.

From Figure 3, it is evident that the Ozawa plot shows a series of straight lines. From Table 2, coefficients of determination R^2^ all are greater than 0.99. The results show that the Ozawa method is successful for PEG and PEG block in PEG-PLLA using methods 1 and 2.

In fact, for many copolymers, homopolymers, and blinks, Ozawa theory has been shown to be inadequate to describe their non-isothermal crystallization kinetics [25,28,29,30]. They argued that the Ozawa method ignored secondary crystallization, resulting from spherulite impingement and restricted effects, and other reasons, the quasi-isothermal nature of the treatment. In general, primary crystallization and secondary crystallization are two kinds of crystallization processes of crystalline polymers. Ozawa believed that the effect of secondary crystallization could be negligible and it occurred in the later stage in the cooling process. However, there are many factors affecting secondary crystallization, such as cooling rate, which cannot be ignored for some polymer systems. Therefore, this factor should be taken into account when the Ozawa equation is used to analyze the kinetics of non-isothermal crystallization.

The researchers mentioned above all used a relatively low cooling rate in the cooling crystallization process (the fastest is generally 40 °C/min), in which case secondary crystallization occurs very easily and cannot be ignored. As shown in Figure 3, the degree of the data point fitting is very good when we use the fast cooling rate to study, which confirms the secondary crystallization is inhibited under this condition. According to Table 2, $\bar{q}$ is equal to 1.829 in (a) and is equal to 2.491 in (b), it shows that PEG and the PEG block in PEG-PLLA during the crystallization process differed by about one dimension under the same cooling method, and further illustrates that the first crystallized PLLA block have an effect of the crystallization behavior on the later crystallized PEG block. Specifically, the crystal of the PLLA block provides nucleating sites for the crystallization of the PEG block and can be regarded as heterogeneous nucleation to a certain extent [7,22]. The difference of $\bar{q}$ between (b) and (c) means that inhibition of crystallization in the PLLA block has a significant effect on subsequent crystallization. $\bar{q}$ in (c) equals 2.057, between 1.829 and 2.491, which shows that the method of the PEG block and PLLA block crystallized together is intermediate between crystallized PEG and the PEG block and PLLA block crystallized in turn. According to the Ozawa method, $\bar{q}$ equals 2.057 means that the method of the PEG block and PLLA block crystallized together corresponds to one-dimensional growth. Additionally, this reflects from the side that between the crystallization region of the PLLA block and PEG block there exists a certain separation.

The relationship between lnK(*T*) and temperature *T* is shown in Figure 4. The fitting results coefficients of determination R^2^ of the three groups of data (a), (b) and (c) are all greater than 0.92. K(*T*) is the cooling function, and it only varies as a function of temperature. For a given temperature, it depends on the nucleation process of the crystalline entities (homogeneous or heterogeneous), on growth pattern and on growth rate; therefore, different crystallization mechanism has certain influence on it. lnK(*T*) is supposed to be a linear function of the temperature. Like the results obtained by other researchers [31,32], we have the same result that lnK(*T*) is a linear function of temperature. However, it must be considered that the values of K(*T*), in non-isothermal crystallization, do not have the same physical significance as in isothermal crystallization, because the temperature is changing constantly under non-isothermal conditions [10]. This affects the rate of nuclear formation and crystal growth because they are temperature dependent.

### 3.3. Crystallization Activation Energy (ΔE)

Considering the change of peak temperature *T*_max_ with cooling rate, Kissinger derived the Kissinger equation from the Arrhenius equation, with the following form [21]:(4)dlnaTmax2d1Tmax=−ΔER
where *R* is the gas constant (8.314 J/(mol·K)), *ΔE* is the activation energy. According to the results in Table 1, the curves of ln(*a*/$T_{max}^{2}$) and 1/*T*_max_ are drawn, as shown in Figure 5. The crystallization activation energy can be obtained from the slope, i.e., ΔE=−R×slope. The Kissinger plot gave a series of parallel lines with a slope and an intercept of function.

The fitting results of the Kissinger equation are listed in Table 3. It can be seen from Table 3 that the determination coefficients R^2^ of the fitting results are all greater than 0.99. Thus, it is evident that the Kissinger equation does adequately describe the non-isothermal crystallization kinetics of PEG and PEG-PLLA. The crystallization activation energies of (a), (b) and (c) are −62.52, −59.05 and −60.71 kJ/mol, respectively. Negative values only indicated that the crystallization process is an exothermic reaction [33], which is consistent with Figure 2, while the absolute value of activation energy represented crystallization capacity, which is (a), (c) and (b) in order. The activation energy of (b) and (c) are smaller than that of (a); that is, the crystallization ability of PEG-PLLA is stronger than that of PEG, which is put down to the influence of PLLA block. Because the activation energy of (b) is less than that of (c), the crystallization ability of PEG block and PLLA block crystallized in sequence is stronger than that of the PEG block and PLLA block crystallized together, which is attributed to the fact that the first crystallized PLLA block promoted the crystallization of PEG block, which is consistent with the results discussed above. In addition, Yang, J.L. et al. [7] measured the absolute value of activation energy of PEG and PEG-PLLA by ordinary DSC, which were greater than 150 kJ/mol, which were far greater than our measured values. This implies that the crystallization capacity of PEG and PEG-PLLA is stronger at fast cooling rates than at slow cooling rates. It can be explained by the fact that crystallization occurred at very low temperatures at fast cooling rates. Then, at lower temperatures, the nucleation rate goes up.

### 3.4. Analysis of Crystallization Rate

The bulk crystallization process of macromolecules is phenomenologically divided into two steps: nucleation and growth. The crystallization process of polymer materials, especially the nucleation induction period, is often very long, and crystallization half time, which the relative degree of crystallinity reaches 50%, is usually used to reflect the total crystallization rate [18,25]. In addition to the activation energy of the Kissinger equation, which reflected the crystallization capacity, we adopted crystallization half time to represent the total crystallization rate, and that is used to describe the crystallization difficulty in the whole process. During the non-isothermal crystallization process, the relation between crystallization half time *t*_1/2_ and the corresponding temperature *T*_1/2_ is
(5)$$t_{1/2}=\frac{T_{1/2}-T_{0}}{a}$$
where *T*_1/2_ is the temperature at half crystallization, *T*_0_ is the initial crystallization temperature, and *a* is the cooling rate. Based on Equation (5), we can calculate crystallization half time at different cooling rates as shown in Table 4. For visual comparison, their variation with cooling rates is shown in Figure 6.

It can be seen that the higher the cooling rate, the shorter the time to complete half of the crystallization; that is, the faster the total crystallization rate. In addition, obviously, it can be seen from these data that the crystallization process has a strong dependence on the cooling rate. With the increase in cooling rate, the crystallization half time becomes closer. When the cooling rate is 400 K/s, the relative crystallinity of 50% can be reached in almost the same time. For PEG and the PEG block in PEG-PLLA under the same cooling method, the crystallization half time of PEG is always smaller than that of PEG in PEG-PLLA, which indicates that the total crystallization rate of PEG in PEG-PLLA is slower. In combination with the above discussion, although crystallization of the PLLA block provides heterogeneous nucleation conditions for the PEG block to a certain extent, it does not shorten the time of the whole crystallization process, because the whole crystallization process includes nucleation and growth, which also reflects the complexity of the crystallization process. Moreover, for the two cooling methods of PEG-PLLA, the crystallization half time of method 1 is always greater than that of method 2. It means that when the PLLA block crystallizes first and the PEG block crystallizes later, the PEG block crystallizes more slowly than when PEG and PLLA crystallize together. This results from the confined crystallization effect of the PLLA block [10].

## 4. Conclusions

The non-isothermal crystallization kinetics of double-crystallizable copolymer PEG-PLLA was investigated and compared with that of PEG homopolymers based on the large crystallization temperature difference between the PEG block and PLLA block in PEG-PLLA and the fast cooling rate provided by FSC. The crystallization process is highly dependent on the cooling rate, and with the increase in the cooling rate, the non-isothermal recrystallization curves tend to flatten during the crystallization process and the crystallization peaks become wider and shift to the lower temperature region. When using rapid cooling rates, the Ozawa theory is a perfect success because secondary crystallization is inhibited. According to Ozawa method, the crystallization of PLLA block provides nucleating sites for the crystallization of PEG block and can be regarded as heterogeneous nucleation to some extent, while the method of the PEG block and PLLA block crystallized together corresponds to a one-dimensional growth, which reflects that there is a certain separation between the crystallization regions of the PEG block and PLLA block. The relative crystallinity *X*_max_ corresponding to the highest peak in the crystallization curves and the crystallization activation energy calculated by the Kissinger equation indicates that the first crystallized PLLA block promotes the crystallization of the PEG block. Crystallization half time indicates at the same time that, although crystallization of the PLLA block provides heterogeneous nucleation conditions for the PEG block to a certain extent, it does not shorten the time of the whole crystallization process because of the complexity of the whole crystallization process including nucleation and growth. In short, due to the fast cooling rate of FSC and the new cooling method, we obtained some different results from previous studies and new understanding about PEG-PLLA.

## Figures and Tables

**Figure 1 polymers-13-01156-f001:**
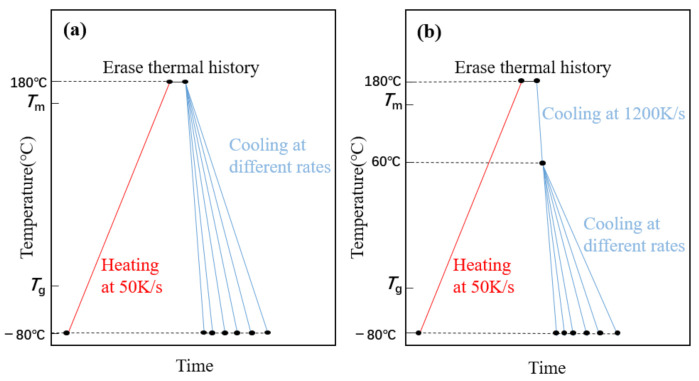
(**a**) Time—temperature profile of method 1. (**b**) Time—temperature profile of method 2 (the highest temperature is 180 °C for 1 s to erase the thermal history, the middle temperature is 60 °C and the lowest temperature is −80 °C).

**Figure 2 polymers-13-01156-f002:**
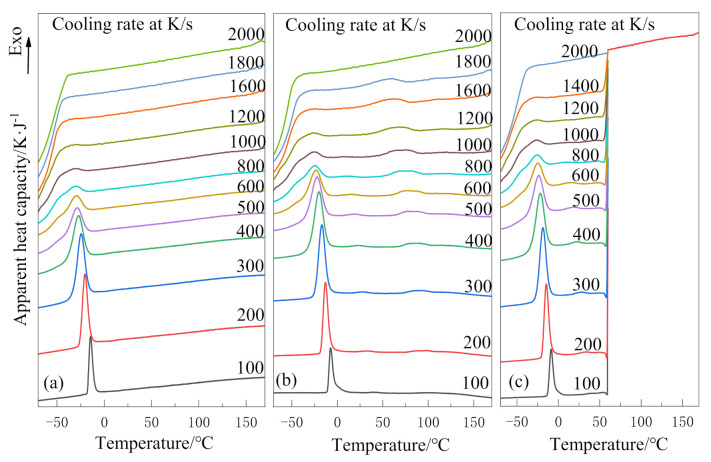
Non-isothermal crystallization curves at different cooling rates: (**a**) poly(ethylene glycol) (PEG) in method 1, (**b**) poly(ethylene glycol)–poly(l-lactide) diblock copolymer(PEG-PLLA) in method 1 and (**c**) PEG-PLLA in method 2.

**Figure 3 polymers-13-01156-f003:**
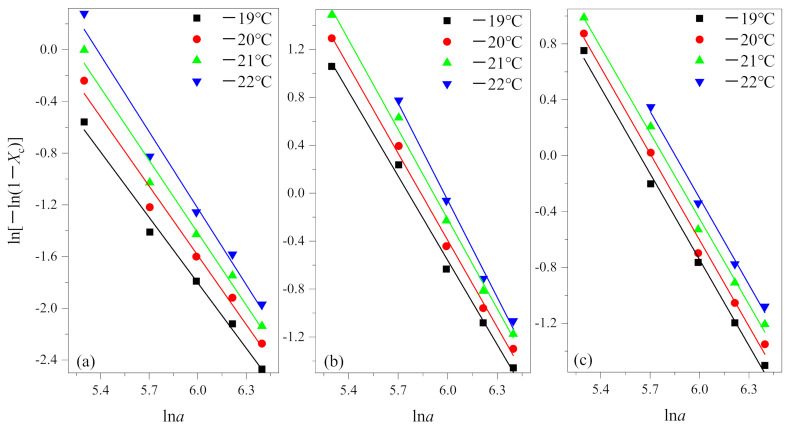
Plots of ln[−ln(1−*X*_c_)] versus ln*a* for non-isothermal crystallization of (**a**) PEG in method 1, (**b**) PEG-PLLA in method 1 and (**c**) PEG-PLLA in method 2.

**Figure 4 polymers-13-01156-f004:**
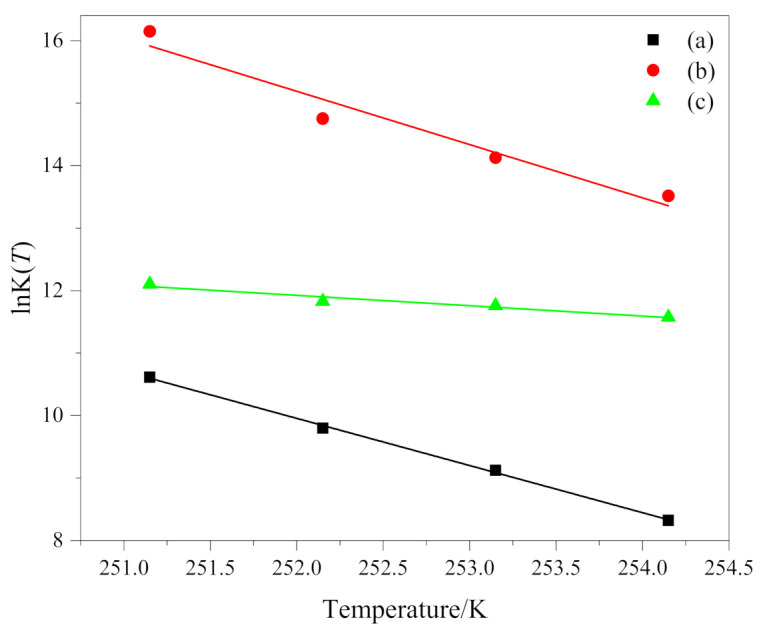
lnK(*T*) as a function of temperature *T* of (**a**) PEG in method 1, (**b**) PEG-PLLA in method 1 and (**c**) PEG-PLLA in method 2.

**Figure 5 polymers-13-01156-f005:**
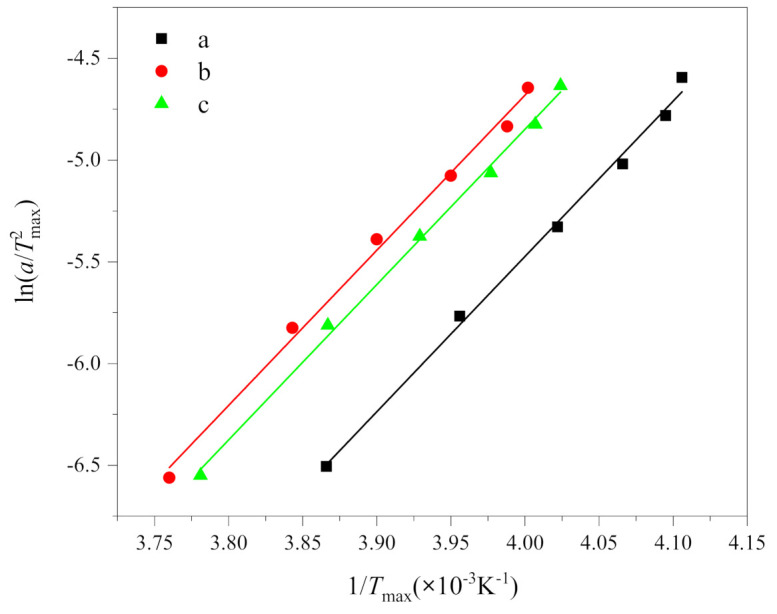
Plots of ln(*a*/$T_{max}^{2}$) versus 1/*T*_max_ according to the Kissinger equation of (**a**) PEG in method 1, (**b**) PEG in PEG-PLLA in method 1 and (**c**) PEG in PEG-PLLA in method 2.

**Figure 6 polymers-13-01156-f006:**
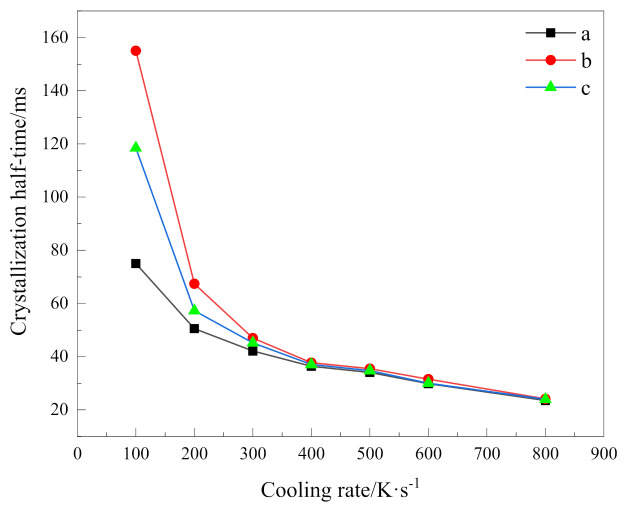
Crystallization half time versus cooling rate of (**a**)PEG in method 1, (**b**)PEG-PLLA in method 1 and (**c**)PEG-PLLA in method 2.

**Table 1 polymers-13-01156-t001:** Characteristic parameters of PEG and PEG-PLLA during the non-isothermal crystallization process: (a) PEG in method 1, (b) PEG-PLLA in method 1 and (c) PEG-PLLA in method 2.

**Cooling Rate (K/s)**	a			b			c		
	*T*_s_ (°C)	*T*_max_ (°C)	*X*_max_ (%)	*T*_s_ (°C)	*T*_max_ (°C)	*X*_max_ (%)	*T*_s_ (°C)	*T*_max_ (°C)	*X*_max_ (%)
100	−6.40	−14.50	0.596	9.30	−7.20	0.662	3.90	−8.70	0.641
200	−9.70	−20.40	0.550	1.38	−12.94	0.570	−2.40	−14.58	0.591
300	−11.30	−24.54	0.481	−3.00	−16.77	0.518	−6.00	−18.66	0.520
400	−13.32	−27.20	0.450	−6.30	−19.96	0.470	−7.68	−21.72	0.442
500	−14.15	−28.96	0.438	−8.05	−22.40	0.451	−8.25	−23.60	0.358
600	−16.10	−29.59	0.331	−10.20	−23.28	0.382	−11.30	−24.64	0.340
800	−19.76	−30.24	0.284	−12.15	−24.20	0.306	−13.52	−24.80	0.316
**/**	${\bar{X}}_{\max}$ = 0.447			${\bar{X}}_{\max}$ = 0.480			${\bar{X}}_{\max}$ = 0.458		

**Table 2 polymers-13-01156-t002:** The non-isothermal crystallization kinetics parameters based on the Ozawa method for (a) PEG in method 1, (b) PEG-PLLA in method 1 and (c) PEG-PLLA in method 2.

**Temperature/°C**	a			b			c		
	R^2^	*q*	lnK(*T*)	R^2^	*q*	lnK(*T*)	R^2^	*q*	lnK(*T*)
−19	0.9972	1.688	8.3218	0.9949	2.346	13.5153	0.9949	2.053	11.5732
−20	0.9937	1.786	9.1223	0.9965	2.420	14.1260	0.9922	2.061	11.7593
−21	0.9928	1.869	9.7976	0.9957	2.495	14.7527	0.9935	2.047	11.8273
−22	0.9917	1.973	10.6133	0.9941	2.702	16.1496	0.9904	2.068	12.1035
/	$\bar{q}$ = 1.829			$\bar{q}$ = 2.491			$\bar{q}$ = 2.057		

**Table 3 polymers-13-01156-t003:** The fitting results of Kissinger equation (a) PEG in method 1, (b) PEG-PLLA in method 1 and (c) PEG-PLLA in method 2.

	R2	Slope (×103)	*Δ**E* (kJ/mol)
a	0.9946	7.52	−62.52
b	0.9937	7.10	−59.03
c	0.9964	7.30	−60.69

**Table 4 polymers-13-01156-t004:** Crystallization half time for different cooling rates of (a) PEG in method 1, (b) PEG-PLLA in method 1 and (c) PEG-PLLA in method 2.

Cooling Rate (K/s)		100	200	300	400	500	600	800
Crystallization half-time (ms)	a	75.00	50.50	42.10	36.40	34.10	29.83	23.50
	b	155.00	67.40	47.00	37.70	35.50	31.53	24.14
	c	118.50	57.25	45.20	37.05	34.70	30.03	23.93

## Data Availability

Not applicable.

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
