# Peer review of "Non-Isothermal Crystallization Kinetics of Poly(Ethylene Glycol)–Poly(l-Lactide) Diblock Copolymer and Poly(Ethylene Glycol) Homopolymer via Fast-Scan Chip-Calorimeter"

_polymers, 2021, doi:10.3390/polym13071156_

Round 1

Reviewer 1 Report

The experiment is well-designed, and the results and discussion are solid. However, before considering for publication, the following issues showed be addressed.

Page 3, line 106-107, way 1 – cool directly from 180°C to -80°C and not 80°C.

Figure 1 (b) cooling rate at 1200K/s between 180°C and 60°C, is correct?

Figure 3 correct the temperatures in the graph caption.

Please provide the unit of temperature in Figure 4 and 5

Reviewer 2 Report

 Chen_Non-isothermal Crystallization Kinetics of Poly(ethylene glycol) - Poly(L-lactide) Diblock Copolymer And Poly(ethylene glycol) Homopolymer Via Fast-Scan Chip-Calorimete Author(s): D. Chen et al

In this manuscript, the authors have studied the Fast-Scan Chip-Calorimeter application (FSC) to understand non-isothermal crystallization kinetics for a hybrid polymer made of PEG block and PLLA block. This is an interesting work, and understanding the hybrid polymeric materials' crystallization is very challenging as each material depending on the molecular structure, follows a different route to crystallization. Overall, I think it is a very good manuscript and written in clear language and should be published subject to some minor modifications discussed below.

- The interactions of the two polymers are a significant factor in the crystallization of the PEG-PLLA system. This is similar to the situation where polymers are loaded with particles to make nanocomposite materials. I think some mention of nanocomposites' crystallization and hybrid polymers in the introduction part would widen the relevance to these systems [J. Chem. Phys., 2017, 147, 020901; Nanomaterials, 2019, 9 (10), 1472; Nanoscale Advances., 2019,1, 4704-4721].

-In Equation 2,3, and elsewhere in the plots and tables, the natural logarithm is shown by "lg".This should be changed to "ln", which is the correct notation.

-The authors show the secondary crystallization is inhibited in fast crystallization. This is very interesting results.

They have demonstrated differences in PEG and PEG-PLLA crystallization, depending on the thermal history and temperatures. However, only one PEG-PLLA material is used with 5000/2000 molecular weight. However, there is no mention of the weight percentage for each block. That would be helpful to have this data. Can the authors also comment on how the system crystallization behaviour might change if the different components of PEG-PLLA were used? (same molecular weights, but different weight percentage ratios).
